# Soft Computing Techniques for Laser-Induced Surface Wettability Control

**DOI:** 10.3390/ma14092379

**Published:** 2021-05-03

**Authors:** Gennaro Salvatore Ponticelli, Flaviana Tagliaferri, Silvio Genna, Simone Venettacci, Oliviero Giannini, Stefano Guarino

**Affiliations:** 1Faculty of Engineering, University of Rome ‘Niccolò Cusano’, Via Don Carlo Gnocchi 3, 00166 Rome, Italy; flaviana.tagliaferri@unicusano.it (F.T.); simone.venettacci@unicusano.it (S.V.); oliviero.giannini@unicusano.it (O.G.); 2Department of Enterprise Engineering, University of Rome ‘Tor Vergata’, Via del Politecnico 1, 00133 Rome, Italy; silvio.genna@uniroma2.it

**Keywords:** soft computing, fuzzy logic, genetic algorithms, laser texturing, wettability.

## Abstract

Making decisions and deducing control actions in manufacturing environments requires considering many uncertainties. The ability of fuzzy logic to incorporate imperfect information into a decision model has made it suitable for the optimization of both productivity and final quality. In laser surface texturing for wettability control, in fact, these aspects are governed by a complex interaction of many process parameters, ranging from those connected with the laser source to those concerning the properties of the processed material. The proposed fuzzy-based decision approach overcomes this difficulty by taking into account both the random error, associated with the process variability, and the systematic error, due to the modelling assumptions, and propagating such sources of uncertainties at the input level to the output one. In this work, the laser surface texturing was carried out with a nanosecond-pulsed laser on the surfaces of AISI 304 samples, changing the laser scanning speed, the hatch distance, the number of repetitions, and the scanning pattern. A significant change of the contact angle in the range 24–121° is observed due to the produced textures. The fuzzy maps highlight the inherent uncertainty due to both the laser texturing process and the developed model.

## 1. Introduction

During the last decades, research in surface engineering has been mainly focused on the development of innovative technologies for the functionalization of components’ surface [1]. In fact, it is recognized that the surface characteristics can control the specific functional performance requested [2]. However, the relationship between surface properties, technology, and application field, requires a deeper comprehension.

To date, many technologies and strategies have been adopted and/or developed to fabricate functionalized surfaces [3], especially on commercial metals, given the large number of applications, both industrial and civil [4]. Among them, surface texturing techniques are considered the most promising due to their low processing costs and adequate production rate [5,6,7]. In particular, laser surface texturing is appeared to be a valid solution to produce tailor-made topographies on the surface of the components by precisely controlling the size and the shape of the texture features [8,9]. Nowadays lasers have demonstrated to be flexible, selective, accurate and efficient technology, that can be successfully used for a wide variety of processes, from cutting [10], to welding [11], to 2D and 3D metal sheet forming [12], to surface heat treatments [13] and for a wide range of materials, from polymers [14] and metals [15], to ceramics [16] and composites [17]. However, the use of laser technology for surface texturing is limited by two main aspects, i.e., the relatively low texturing speed [18] and the residual stress due to the thermal effect of the laser treatment [19]. The application of ultrafast lasers, from picosecond to femtosecond pulsed lasers, can mitigate these limitations [20,21], but increases the cost and the complexity of the laser system. On the other hand, nanosecond pulsed lasers can lead to similar results with a less complex system and a less expensive process, therefore suitable for industrial applications [22].

Among the potential applications of laser technology for surface engineering that have been investigated so far, many of them have been devoted to the laser texturing of metals to produce surface patterns for the control of their wettability. This is due to the many fields in which they can be applied, i.e., chemical sensing and painting [23], biomedicine [24], microfluidics [25], heat transfer [26], anti-corrosion [27], aeronautics [28], and so forth. Therefore, controlling laser-induced surface wettability represent an attractive topic from both the academic and the industrial point of view. However, obtaining the required textures for wettability control, with lasers as well as for any other surface texturing technique, is critical due to the transition between hydrophilicity and hydrophobicity that the surface can experience even for small changes in the surface pattern and/or surface chemistry [29].

Therefore, the definition of the optimal process window in order to reach the requested wetting behavior is still a partially solved task. In fact, during the laser treatment, the process parameters, i.e., those related to the laser source and to the material of the sample under processing, interact with each other in a complex manner, thus complicating the decision-making process [30]. This is of crucial relevance especially for those manufacturing processes deemed to be “advanced” and/or “innovative”. In fact, most of them lack clear and well-defined guidelines in terms of operational parameters and resulting properties, as it is for the laser texturing for surface wettability control [29]. This can cause uncertainty during the manufacturing process, therefore lengthening the production time and increasing the production costs. In this context, research in laser surface texturing processes optimization plays a critical role in advancing such technology and its knowledge.

To advance these technologies, and for their knowledge to be taken up by industry, researchers and manufacturers follow two approaches, i.e., run experiments and/or develop predictive models. During experiments, the parameters are adjusted empirically, i.e., through trial and error, to obtain the quality desired [31,32]. However, this approach can be very expensive, can involve a lot of human resources, can take a long time, remaining at the end unable to provide an adequate estimate of the final quality. For this reason, computational methods for manufacturing optimization problems appear to be a potential solution to support and improve the decision-making process. In fact, in several studies [32,33,34,35], both analytical and/or numerical models were developed with the aim to find the optimal relationship between the process parameters and the process outcomes. However, it is worth noting that each method is limited to the context of the relative study, therefore, a deeper investigation is required for more general applications.

The aim should be the development of a physical model able to simulate accurately the entire process, but such computational simulations are very complex, requiring many different models and very often a high computational cost [36]. For this reason, analytical and empirical modelling can be considered suitable solutions. In particular, the analytical models are obtained through a mathematical analysis of the physical laws and the relevant physical processes involved, but they are limited by the underlying assumptions that can be sometimes very restrictive. While the empirical models, which are the outcome of several experiments, require minimum effort, and once chosen, they can be verified by further tests and used to find out the relationship between the operational parameters able to guarantee the optimal process outputs [37].

Therefore, empirical modelling can be considered a fast and easy solution for establishing a relationship between inputs and outputs of manufacturing processes. However, choosing the best model is not a straightforward task when there are many input variables and the data set adopted for the development of the empirical model is affected by dispersion, which is typically due to the process variability. In this context, genetic algorithms have been successfully implemented for optimization purposes in a robust and efficient way [38,39,40]. However, their effectiveness can be compromised by the fact that the variability of the process together with the approximation introduced in the model lead to a discrepancy between the results of the model and the process outcomes [41]. In particular, the uncertainty due to the process variability is usually aleatoric and can be modelled with stochastic methods, while, the other source of uncertainty is a systematic error for which statistics cannot be used fruitfully. In this context, fuzzy arithmetic represents a valuable solution able to model at the same time both the aleatoric (i.e., statistical) and the epistemic (i.e., systematic) uncertainties [42], and propagate them at the input level to the output quantities [43,44]. The use of soft computing techniques, i.e., fuzzy logic and genetic algorithms, represents therefore a practical way for developing nonlinear control systems which are difficult to design using traditional methods [45].

This work aims to propose a fuzzy-based approach for the optimization of the laser surface texturing of stainless steel for wettability control. The activity consisted of two steps: a first experimental campaign aimed at identifying the effect of the process parameters, i.e., laser scanning speed, hatch distance, and number of repetitions, on the contact angle and the surface roughness [35]; then, based on the acquired data, a genetic algorithm-optimized fuzzy (GAF) regression model was developed and successfully applied to describe the inherent uncertainties related to the investigated process. In particular, the use of the genetic algorithm, previously adopted for the definition of the optimal empirical regression model [35], now is used to find the optimal shape of the membership function of the fuzzy numbers. Then, the input uncertainty, represented through the fuzzy regression coefficients, is propagated to the output variables using the transformation method [46]. The process maps obtained by applying the GAF model are used to select the optimal process parameters able to guarantee the most performing mechanical properties, providing, as additional information, how much the uncertainties introduced by the model and inherently related to the process vary by changing the process parameters themselves.

## 2. Materials and Methods

### 2.1. Experimental Investigation

The starting material is a cold-rolled AISI 304 stainless steel sheet with a thickness of 1 mm and average surface roughness (Ra) of ~0.082 µm. From this, 27 plates with dimension 75 × 25 mm^2^ were cut for the experimental tests. Table 1 and Table 2 report the chemical composition and the main mechanical properties, according to the technical data sheet provided by the producer. On every plate, four squared areas, 225 mm^2^ each, were texturized with a 30 W Q-switched ns-pulsed Yb:YAG fibre laser (YLP-RA30-1-50-20-20, IPG, Oxford, MA, USA), whose main characteristics are reported in Table 3.

The main control parameters of the laser texturing process that can be managed are the laser scanning speed, the hatch distance (i.e., the distance between the centres of two consecutive scan lines), the number of repetitions (i.e., the number of times a pattern is repeated on the sample), the scanning pattern, and the pulse frequency. The experimental investigation was therefore carried out on the basis of a multilevel factorial design [47], as reported in Table 4. It is worth noting that the process parameters and their values have been evaluated through a preliminary investigation aimed at reducing the range within which they could vary. As can be seen in the latter table, the laser scanning speed, the hatch distance, the number of repetitions, and the scanning pattern were varied while controlling the pulse frequency at 60 kHz, which allows achieving the maximum pulse energy and pulse power of 0.5 mJ and 10 kW respectively. Each experimental condition, i.e., 36 (Table 5), was replicated three times for a total of 108 tests. The scanning pattern consisted of parallel lines along with two (+90°, −90°) and four (+90°, −90°, +45°, –45°) directions. Therefore, the number of repetitions is an integer multiple of two. Moreover, each sample was cleaned with acetone in an ultrasonic bath for 2 min both before and after the laser processing. The sequence of the experimental tests was performed randomly, but by the same operator under the same conditions in order to reduce any additional disturbance.

The quality outputs investigated in this research work [35] were the apparent contact angle (ϑ) and the surface roughness in terms of arithmetic mean roughness profile (Ra) and developed surface area ratio ($r_{f}$). In fact, the apparent contact angle is suggested to be of major interest for wettability control [48], together with Ra and $r_{f}$, which are used to express the increment of the interfacial surface area and the projected one that can promote or hinder the penetration of the liquid within the surface asperities thus enabling the establishment of the hydrophilic or hydrophobic behavior [49]. Thus, for $r_{f}$ equal to 1 the surface is flat, while when $r_{f}$ is greater than 1 means that the texture contributes with an additional surface area. Consequently, the resulting heterogeneity of the textured surface could lead to the formation of regions which are fully penetrated by the liquid and others with partial or no penetration at all. In particular, it is possible to distinguish four wettability states, as shown in Figure 1: (i) superhydrophilic, when there is no equilibrium and the penetration front of the drop spreads all over the substrate surface forming a liquid film, i.e., the contact angle is lower than 5°; (ii) hydrophilic, if the liquid completely wets the textured surface of the sample with a contact angle lower than 90°; (iii) hydrophobic, if the air pockets let the drop to partially siting on the substrate surface and the contact angle ranges between 90° and 150°; (iv) superhydrophobic, when the drop of the liquid sit on the textured surface due to the air trapped between the asperities, leading to contact angles greater than 150°.

The apparent contact angle (ϑ) was calculated according to Equation (1) [50] by measuring the height of the droplet (h) and the radius projected on the substrate (r), shown in Figure 2a. The liquid adopted is distilled water at room temperature (i.e., 22 °C). A pipette with a volume of 2–20 µL ± 0.010 µL (Pipetman P20, Gilson Italy, Milan, Italy) was used to deposit a 10 μL drop on the textured surface of each sample. The height and the radius of the droplets were measured by analysing the optical images captured by a 3D digital microscope (KH-8700, Hirox, Tokyo, Japan). The setup for the measurement of the apparent contact angle is shown in Figure 2b. While the surface topography was characterized by using a high-resolution 3D profilometer (Talysurf CLI 2000, Leicester, UK) for the measurement of the average surface roughness (Ra) according to the standard UNI EN ISO 4288:2000, and optically inspected with the 3D digital microscope.
(1)ϑ=2arctan(hr).

### 2.2. Computational Modelling

In previous works [35,40], an optimization method based on genetic algorithms has been proposed with the aim to identify the optimal empirical model for the laser texturing process. Despite the ability of the optimised model to reproduce the measured data acceptably, the resulting discrepancy can be addressed to two sources of error, i.e., the simplification introduced by the model, which is of epistemic nature, and the dispersion due to the variability of the process, which is aleatoric. While the latter source of uncertainty can be modelled with stochastic methods, the other one is a systematic error for which statistics does not provide a useful tool. In this context, fuzzy arithmetic represents a valuable solution able to model at the same time both the epistemic and aleatoric uncertainties [42], and propagate them from the input level to the output quantities [43]. The use of soft computing techniques, i.e., fuzzy logic and genetic algorithms, represents therefore a practical way for developing nonlinear control systems which are difficult to design using traditional methods [45].

The proposed approach consists of the development of a fuzzy regression model starting from the regression model previously optimized through the application of the genetic algorithm, described by Equation (2):(2)$$\begin{array}{c} y\left(S_{L},d_{H},R_{n},S_{P}\right)=a\cdot S_{L}{}^{\alpha_{1}}d_{H}{}^{\beta_{1}}R_{n}{}^{\gamma_{1}}S_{P}{}^{\delta_{1}}+b\cdot S_{L}{}^{\alpha_{2}}d_{H}{}^{\beta_{2}}R_{n}{}^{\gamma_{2}}S_{P}{}^{\delta_{2}}+c\cdot \\ S_{L}{}^{\alpha_{3}}d_{H}{}^{\beta_{3}}R_{n}{}^{\gamma_{3}}S_{P}{}^{\delta_{3}}+d\cdot S_{L}{}^{\alpha_{4}}d_{H}{}^{\beta_{4}}R_{n}{}^{\gamma_{4}}S_{P}{}^{\delta_{4}}+e \end{array}$$
where $y\left(S_{L},d_{H},R_{n},S_{P}\right)$ is the output variable expressed as a function of the input parameters, i.e., laser scanning speed ($S_{L}$), hatch distance ($d_{H}$), number of repetitions ($R_{n}$), scanning pattern ($S_{P}$); while a, b, c, d, e, are the empirical regression coefficients determined by linear regression analysis based on the whole experimental data set. In particular, in order to match the simplest linear model that can be built by considering the effect of the single parameters, the proposed model is constituted by a number of terms equal to the number of parameters, i.e., four, plus a constant term, for a total of five terms. Each term is dependent on all the parameters to the powers $\alpha_{i}$, $\beta_{i}$, $\gamma_{i}$, $\delta_{i}$, for $S_{L}$, $d_{H}$, $R_{n}$, $S_{P}$, respectively, with i = 1, 2, 3, 4, for the first, second, third, and fourth term. The powers were chosen between −1, −0.5, 0, 0.5, 1.

Starting from the empirical model so optimized (from Equation (2)), i.e., once found $\alpha_{i}^{opt}$, $\beta_{i}^{opt}$, $\gamma_{i}^{opt}$, $\delta_{i}^{opt}$, for each output variable, a different fuzzy regression model can be written, as described by Equation (3):(3)$$\begin{array}{c} y^{*}\left(S_{L},d_{H},R_{n},S_{P}\right)=a^{*}\cdot S_{L}{}^{\alpha_{1}^{opt}}d_{H}{}^{\beta_{i}^{opt}}R_{n}{}^{\gamma_{1}^{opt}}S_{P}{}^{\delta_{1}^{opt}}+b^{*}\cdot S_{L}{}^{\alpha_{2}^{opt}}d_{H}{}^{\beta_{2}^{opt}}R_{n}{}^{\gamma_{2}^{opt}}S_{P}{}^{\delta_{2}^{opt}}+c^{*}\cdot \\ S_{L}{}^{\alpha_{3}^{opt}}d_{H}{}^{\beta_{3}^{opt}}R_{n}{}^{\gamma_{3}^{opt}}S_{P}{}^{\delta_{3}^{opt}}+d^{*}\cdot S_{L}{}^{\alpha_{4}^{opt}}d_{H}{}^{\beta_{4}^{opt}}R_{n}{}^{\gamma_{4}^{opt}}S_{P}{}^{\delta_{4}^{opt}}+e^{*} \end{array}$$
where $y^{*}\left(S_{L},d_{H},R_{n},S_{P}\right)$ is the fuzzy function determined through the Transformation Method [46], while $a^{*}$, $b^{*}$, $c^{*}$, $d^{*}$, $e^{*}$, are fuzzy numbers. Such an approach is proved to be able to propagate both sources of uncertainty, i.e., the random one due to the process variability and the systematic one due to the simplification introduced by the model itself, to the output qualities [51,52]. This is carried out by implementing the fuzzy arithmetic as a series of interval computations, i.e., α-cut strategy, which are numerically solved by sampling the interval based on a defined schema. In the previous works, the membership functions of the fuzzy regression coefficients were represented with a triangular shape, as shown in Figure 3a. However, due to the inherent linearity of the triangular fuzzy number, although a decrease in the level of uncertainty, i.e., an increase in the membership function $\mu \left(x_{i}\right)$, implies considering a smaller number of data with higher membership level, the relation between this number and the corresponding membership level is not strictly defined. For this reason, in this work, such an inherently nonlinear relation is evaluated through the implementation of a genetic algorithm. The objective is therefore the identification of nonlinear fuzzy numbers for the regression model, shown in Figure 3b, with adaptive supports capable of including, at a specific α-level, a specific number of data points.

As shown in Figure 3, the fuzzy number is defined through three values, i.e., the modal value $m_{i}$, the lower bound $l_{i}$ and the upper bound $u_{i}$. The first is the value of the coefficient calculated with the standard linear regression, while the other two represent the bounds of the fuzzy support. The membership function μ is represented as a grey shaded area varying from white, i.e., μ = 0 and α-level = 1, to black, i.e., *µ* = 1 at the last α-level. Therefore, moving from a triangular representation of the fuzzy number to a nonlinear one, the relation between represented data and membership level is more rigorous: e.g., 100% of data at *µ* = 0; 90% of data at *µ* = 0.1; up to 0% at *µ* = 1, which is the crisp regression model obtained from the experimental data set. Consequently, the greater the membership, the more precise the model, but less descriptive of the relation because a lower number of measured data points fall on.

The use of genetic algorithms is aimed at optimizing the interval at each α-level j. To this end, the genetic algorithm performs several constrained optimizations to identify the optimal interval solution that envelops in the output space a defined subset of experimental data points within the minimum hyper-volume of the envelop ($V_{j}$), i.e., the target is the minimization of the fitness function $f_{j}$, defined by Equation (4):(4)$$f_{j}=\min\left(V_{j}\right),$$

The procedure adopted is the same as the optimization of the regression models described in the previous works [35,40], while here the algorithm is iterated for each fuzzy number at each α-level, returning as results the optimal bounds of the fuzzy supports, and the selection operator is performed after the crossover and mutation in order to further increase genetic variability. Figure 4 shows the flowchart of both algorithms and how they are linked to each other. Briefly, The genetic algorithms are computed through four fundamental operations [53]: (i) initialization, with the definition of a set of chromosomes, i.e., powers of the regression model terms first and supports of the fuzzy numbers then. Chromosomes are encoded and represented in terms of strings of bit by means of the binary encoding procedure, which is the most common form of encoding that maximizes the number of exploitable schemas [54]. Then, the fitness functions are applied to evaluate the fitness of this population, and if the stop condition is not met, it evolves into the next generation through the following genetic operators; (ii) crossover, which increases the variability of populations through the exchange of genes between two random chromosomes, i.e., parents, operating randomly on a single point of every chromosome. This allows obtaining different individuals of which the most fitted are kept by the selection operator; (iii) mutation, which introduces random variation in the genome of some individuals. This allows avoiding local convergence of the genetic algorithm, thus promoting diversity and the occurrence of more powerful generations [39]. Moreover, in this case, this operator is used in parallel with the crossover to emphasize the gains on the algorithm performance due to the concurrent application of operators with complementary roles [40]; (iv) selection, which allows transferring a defined number of chromosomes in the population to the next generation. The selection happens through ranking the individuals on the basis of their fitness values, keeping the best half, and eliminating the others. Finally, these steps are implemented in an iterative procedure that continues until the stop condition is met, i.e., a predefined number of generations.

It is worth highlighting here that even if the proposed genetic algorithm is based on the classic method of artificial evolution, which simulates the evolution of living things, the obtained results presented in Section 3.2 should be considered valid only in the solution space defined in this work, since there is no guarantee to reach the same global optimum for different operational conditions or processes in general [53]. In fact, some of the key parameters, i.e., crossover and mutation probability, were chosen in order to guarantee the global convergence of the algorithm in an affordable computational time.

## 3. Results and Discussion

### 3.1. Experimental Findings

In order to consider the wettability states presented in Section 2.1 valid, the drop size should be at least three orders of magnitude greater than the average roughness [55]. In this case, in fact, the surface appears uniform to the drop, and the resulting contact angles can be considered as the most stable [56]. In general, the roughness (Ra) ranged between 1.032 µm and 10.238 µm, while the diameter of the droplets at the interface with the sample surface, Figure 2a, between 1.968 mm and 6.514 mm, thus satisfying the mentioned condition.

Figure 5 shows the experimental results in terms of contact angle (ϑ) and developed surface area ratio ($r_{f}$), and how they are affected by the operational process parameters varied in this investigation, i.e., laser scanning speed, hatch distance, number of repetitions, and scanning pattern (see Table 4 and Table 5). In particular, in the latter figure, the black dots represent the mean values of the response variables, while the error bars represent the standard deviations. It is worth noting that only the results related to ϑ and $r_{f}$ are reported, since the average surface roughness Ra gives the same kind of information of $r_{f}$ in terms of effect on the surface quality.

As shown in Figure 5, the contact angle (ϑ) and the developed surface area ratio have a similar trend for three of the four control parameters, i.e., hatch distance, number of repetitions, and scanning pattern, while the trend is the opposite for the laser scanning speed. In particular, a higher hatch distance leads to a higher contact angle and developed surface area ratio, therefore promoting the transition from the hydrophilic to the hydrophobic behaviour. In fact, the greater the hatch distance, i.e., the laser scans separation, the bigger the portion of the sample surface which is not ablated by the laser, therefore the more marked the profiles of peaks and valleys, and the higher the roughness and the developed surface area ratio [49]. Such topography is made of a triple interface between the stainless steel (i.e., solid), of which the sample is made, the air entrapped into the valley (i.e., gaseous), and the distilled water (i.e., liquid) [57]. As a result, due to the reduced surface tension for the high number of peaks and valleys [49], the drop assumes a more spherical shape. As a consequence, also the scanning pattern affects the wettability of the surface by promoting the establishment of a hydrophobic behaviour when the laser scans the surface only along with two directions, i.e., ±90°. In fact, the greater the number of laser scans, the smoother the surface, thus the lower the wettability. The same is true for the number of repetitions, for which the higher the number of times the laser scans the same portion of the sample surface, the deeper the groove, the more enhanced the effect peak-valley, therefore the more the hydrophobic behaviour is likely to take place. On the other hand, an increase in the laser scanning speed results in greater contact angles but a lower developed surface area ratio and therefore lower surface roughness. This can be addressed to the fact that by reducing the laser scanning speed, the time duration of the interaction between the laser and the material surface is greater, therefore removing a greater amount of material [16]. This can lead to the deposition and re-solidification of undesired material from the groove to the surface of the sample, resulting in a reduced average roughness. While, in terms of contact angle, the surface appears to be characterized by a greater number of peaks and valleys and therefore with lower surface tension, thus promoting the hydrophobic behaviour. These findings are supported by the drops’ images recorded for the different process parameters’ combinations, shown in Figure 6.

It is worth noting here that this discussion is limited to the average trend of the effect of the operational parameters, i.e., laser scanning speed ($S_{L}$), hatch distance ($d_{H}$), number of repetitions ($R_{n}$), scanning pattern (SP), on the response variables, i.e., contact angle (ϑ) and developed surface area ratio ($r_{f}$). In order to verify the statistical significance of these results an ANOVA test has been carried out by using the software Minitab. For sake of briefness, the interactions between the parameters were limited to two. Table 6 reports the results obtained for the contact angle, while Table 7 for the developed surface area ratio. As can be seen in the latter, the effect of each process parameter can be considered statistically significant: *p*-value < 0.0.5, and F-value > 3.97 for DoF = 1, F-value > 3.12 for DoF = 2, or F-value > 2.50 for DoF = 4, where DoF is the degrees of freedom. Therefore, the average trends can be considered representative of the variation of both output variables for varying values of the process parameters.

Finally, Figure 7 shows the results of the experimental campaign in terms of wettability states obtained for the different combinations of the control parameters here investigated, i.e., laser scanning speed, hatch distance, number of repetitions, and scanning pattern (see Table 4). As can be seen, only six scenarios are characterized by the hydrophobic behaviour (90° < ϑ < 150°), i.e., 14 to 18 and 35 (Table 5). All of them are obtained by applying the same number of repetitions, i.e., 40, which is the highest adopted (the other is 8), suggesting them to be the factor that affects the most the wettability of the surface. In fact, as previously highlighted, a greater number of repetitions lead to deeper grooves and therefore to a sharper difference between peaks and valleys, as demonstrated by the higher values of the developed surface area ratio. This result is further supported by the values of the hatch distance, i.e., 100 µm and 150 µm, and the ±90° scanning pattern, which, together with the highest values of the scanning speed, leads to the formation of bigger and sharper asperities that allow the drop to partially sitting on the air trapped between the drop itself and the pockets so formed. In particular, Figure 8 shows the surface topography of a sample textured by adopting the parameters of the condition 18, i.e., $S_{L}$ = 1000 mm/s, $d_{H}$ = 150 µm,$\;R_{n}$ = 40, $S_{P}$ = ±90°, compared to the typical surface topography that leads to the hydrophilic state (5° < ϑ < 90°), for example the condition 19, i.e., $S_{L}$ = 400 mm/s, $d_{H}$ = 50 µm,$\;R_{n}$ = 8, $S_{P}$ = ±90°/±45°. Neither the superhydrophilic state (ϑ < 5°) nor the superhydrophobic one (ϑ > 150°) were observed.

### 3.2. Fuzzy Optimization

Figure 5 clearly shows that the discussion made in Section 3.1 about the effect of the process parameters, i.e., $S_{L}$, $d_{H}$, $R_{n}$, $S_{P}$, on the response variables, i.e., ϑ and $r_{f}$, can be considered true for the average trend, highlighted by the black dots. While, if the standard deviation is taken into account, highlighted by the error bars, for many of the experimental conditions, the same conclusions cannot be drawn. This discrepancy is typically attributed to the process variability, thus introducing the first source of uncertainty which is of aleatoric type, so when developing empirical models in order to relate input(s) and output(s), this random error can be modelled with stochastic methods by adding an opportune contribution term within the model. However, the model itself introduces a new source of uncertainty due to the simplification assumptions, which are epistemic and therefore not efficiently describable by statistics [42]. The use of soft computing techniques, fuzzy logic and genetic algorithms, in this case, represents therefore a practical way for developing control systems able to consider both sources of uncertainty at the input level and propagate them at the output level [36].

The first step, before the application of the proposed fuzzy approach, has been the optimization of the regression models for each output variable (i.e., Ra, ϑ, $r_{f}$) through the genetic algorithm developed on the basis of the previous works [35,40]. So, starting from the empirical model described by Equation (2), the optimal values of the powers of the model terms, i.e., $\alpha_{i}^{opt}$, $\beta_{i}^{opt}$, $\gamma_{i}^{opt}$, $\delta_{i}^{opt}$, have been found and applied to evaluate the empirical coefficients ($a_{opt}$, $b_{opt}$, $c_{opt}$, $d_{opt}$, $e_{opt}$) through standard linear regression. The space of the possible powers is [−1, −0.5, 0, 0.5, 1], therefore discrete and constituted by five terms. Consequently, it contains 5^16^ models, where 16 is given by multiplying the number of variables constituting each term with the number of terms, i.e., four in both cases. The constant term is not considered since the powers are fixed to zero. The set of individuals was set at 5000. In general, within these conditions, less than 45 generations were needed to reach the convergence. Therefore, the algorithm solves only 2.25·10^5^ models out of 5^16^ possible. Table 8 reports the obtained values of the powers and the resulting coefficients.

The second step concerned the application of the proposed fuzzy approach starting from the optimal regression models found so far, with the aim to optimize supports of the fuzzy numbers, and therefore their shapes, by using another genetic algorithm-based method able to accommodate a varying number of experimental data points according to the membership level, as described in Section 2.2.

For sake of briefness, in the following, it is reported only the results obtained for the contact angle as a representative case, described by Equation (5), and able to reproduce the measured data with a root mean square error of 8.44% (the mean standard deviation is 8.37%). Starting from the latter equation, the fuzzy numbers $a^{*}$, $b^{*}$, $c^{*}$, $d^{*}$, $e^{*}$, and the fuzzy function $\vartheta^{*}\left(S_{L},d_{H},R_{n},S_{P}\right)$ are described by 6 α-cuts. At each of them, the genetic algorithm evaluates the optimal fuzzy support, while the fuzzy function is computed through the transformation method. For each α-cut, it requires, in a combinatorial scheme, the evaluation of the number of points within the α-cut, i.e., 4, to the power of the number of fuzzy parameters, i.e., 5, leading to a total of 6·4^5^ = 6144 evaluations for each output variable. Figure 9 shows the nonlinear fuzzy numbers identified by applying the developed genetic algorithm. In this figure, the x-axis represents the support of the fuzzy numbers, which are optimized at each α-level, thus obtaining a peculiar shape for each of them, while Figure 10 reports the results of the fuzzy model implementation order for increasing values of the contact angle:(5)ϑ*(SL,dH,Rn,SP)=aopt*·dHRnSL+bopt*·SLSPRn+copt*·SLSPRn+dopt*·SPSLRndH+eopt*

From Figure 9 it is evident that the most influencing factors are those associated with the first and the last coefficients, i.e., $a_{opt}^{*}$ and $e_{opt}^{*}$. The first is associated with the combination of laser scanning speed, hatch distance, and the number of repetitions, while the last is the constant term. The other three terms, i.e., $a_{opt}^{*}$, $b_{opt}^{*}$, $c_{opt}^{*}$, are characterized by very small support almost coincident to the corresponding modal value, which is already reached at the fourth α-level. This suggests that the regression model optimized through the genetic algorithm is still lacking some information about the relation between the input process parameters and the output, i.e., the contact angle in this example. This is due to the simplification introduced by the model itself. However, such uncertainty can be taken into account, with the proposed fuzzy approach, together with the uncertainty associated with the process variability, and propagated to the output qualities through the transformation method. The result is a fuzzy process map in which the extent of the fuzzy bands varies depending on the operational parameters’ combinations used during the experimental tests (see Figure 10). It is worth highlighting that the extent of the uncertainty bands is related to both the accuracy of the regression model and the variability of the process. In particular, the operator is warned of the degree of uncertainty, which is particularly high for the combinations 10, 13, 16, 28, 29, 30, 31, 33, 32, 34, and 35 (see Table 5 for the details). As a result of the genetic algorithm optimization, the fuzzy model represents the optimal “uncertain” description of the relation between the process parameters and the process outcomes, which accounts for all the sources of uncertainty, i.e., process variability, approximation due to the linear regression, and for any measurement error during the characterization of roughness and contact angles. The membership function μ, represented as greyscale, ranges from 0 (i.e., white) to 1 (i.e., black) and describes the degree of belonging of a given sample to the model, i.e., from 100% (at μ = 0), which represents a non-precise model that take into account all the experimental data, to 0% (at μ = 1), that is the crisp regression model and upon which no measured data point falls on.

Moreover, the fuzzy model can be used to identify the most suitable combination of the process parameters in order to satisfy a desired requirement. In this case it has been implemented for the obtainment of a specific wettability state, i.e., hydrophilic (5° < ϑ < 90°) or hydrophobic (90° < ϑ < 150°), which are the only two wettability states observed during the experimental campaign. For sake of briefness, Figure 11 shows the fuzzy inverse maps obtained for the contact angle belonging to the hydrophobic state drawn by considering the combination of two parameters at a time while keeping the others constant, in this case at their maximum levels except for the scanning pattern, i.e., $S_{L}$ = 1000 mm/s, $d_{H}$ = 150 µm, $R_{n}$ = 40, SP = ±90°. Moreover, in the latter, the experimental results are highlighted by red dots and the relative occurrences by green numbers.

From the inspection of Figure 11, if adopted for the control of the surface wettability, the operator is warned of the variability of the process due to the width of the fuzzy bands that suggest the level of uncertainty related to the specific parameters’ combination. For example, Figure 11 suggests that the hydrophobic state can be promoted by adopting a high number of repetitions, greater than 24, while keeping the hatch distance over 75 µm, the laser scanning speed over 600 mm/s, and limiting the scanning pattern to ±90°. In these scenarios, in fact, the fuzzy areas are darker and narrower, thus representing the parameters’ combination characterized by the lowest level of uncertainty. In other words, within these operational conditions, the possibility to obtain the desired quality is the highest. These findings are also supported by the experimental data. In fact, the highest number of occurrences (i.e., 3) falls on the darkest areas of the fuzzy inverse maps. In fact, as reported in Section 3.1, an increasing value of the hatch distance, number of repetitions, and scanning speed, together with limiting the laser scanning to only two directions (i.e., ±90°), results in a higher contact angle. This happens because in such conditions the profiles of peaks and valleys are more marked and therefore characterized by a reduced surface tension that let the drop to assume a more spherical shape [49], which is characteristic of the hydrophobic state.

These results reveal that the proposed approach based on a combination of soft computing techniques, i.e., genetic algorithms and fuzzy logic, in this case, can be considered a useful tool in estimating regression parameters when the experimental data set is characterized by a non-negligible dispersion between the data points, thus making the statistical regression analysis not suitable to suggest a regression model due to the vague relationships among variables and poor model specification.

## 4. Conclusions

The use of soft computing techniques, i.e., fuzzy logic and genetic algorithms, has been demonstrated to represent a practical way for developing nonlinear control systems which are difficult to design using traditional methods due to the sources of uncertainty that affect the process to control. In this study, the proposed approach, aimed at finding the optimal fuzzy regression model through the application of genetic algorithms, has been successfully applied for the control of the laser-induced surface wettability on stainless steel substrates.

First, the experimental campaign was carried out to create the starting data set on which to develop and train the optimized algorithm. To this end, a Q-switched 30 W nanosecond-pulsed Yb:YAG fiber laser system was adopted on AISI 304 stainless steel plates by varying the laser scanning speed, the hatch distance, the number of repetitions and the scanning pattern. The output qualities investigated were the roughness, in terms of average surface profile roughness and developed surface area ratio, and the wettability, in terms of contact angle.

The results showed that within the experimental conditions adopted in this work only two wettability states can be reached, the hydrophilic and the hydrophobic ones, with contact angles in the ranges 5–90° and 90–150° respectively. Among the operational parameters, the number of repetitions and the hatch distance are the most influencing factors. In fact, they greatly affect the geometry of the texture on the sample surface, i.e., the greater the number of repetitions and the hatch distance, the deeper and the larger the grooves, therefore the greater the distance between two consecutive peaks or valleys.

On the basis of the experimental data set so collected, a genetic algorithm-optimized fuzzy regression model has been developed and applied to find the relation between the input parameters, the output qualities, and the uncertainty related to both the process variability and the simplification introduced by the model itself. The optimal model suggested by the genetic algorithm can reproduce the measured data with a mean error of 8.44%. Finally, the fuzzy inverse maps suggest, according to the experimental results, that, in order to promote the hydrophobic state and at the same time ensure the lowest level of uncertainty, more than 24 repetitions, a hatch distance greater than 75 µm, a laser scanning speed over 600 mm/s, and the ±90° scanning pattern should be adopted.

The method here proposed can be considered of a general nature and applicable in the manufacturing environment to any type of process for which it is possible to have a data set on which to develop the model.

## Figures and Tables

**Figure 1 materials-14-02379-f001:**
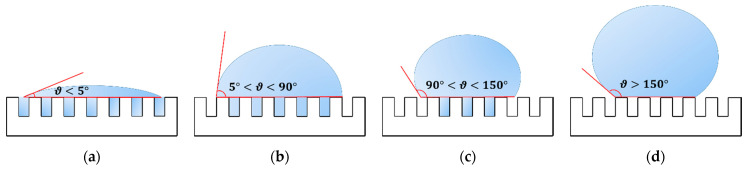
Wettability states: (**a**) superhydrophilic, when there is no equilibrium and the drop spreads all over the substrate surface forming a liquid film, i.e., the contact angle is lower than 5°; (**b**) hydrophilic, if the rough surface of the sample is completely wet by the liquid and the contact angle is lower than 90°; (**c**) hydrophobic, if the air pockets let the drop to partially sit on the substrate surface and the contact angle ranges between 90° and 150°; (**d**) superhydrophobic, when the drop of the liquid sit on the textured surface due to the air trapped between the asperities, leading to contact angles greater than 150°.

**Figure 2 materials-14-02379-f002:**
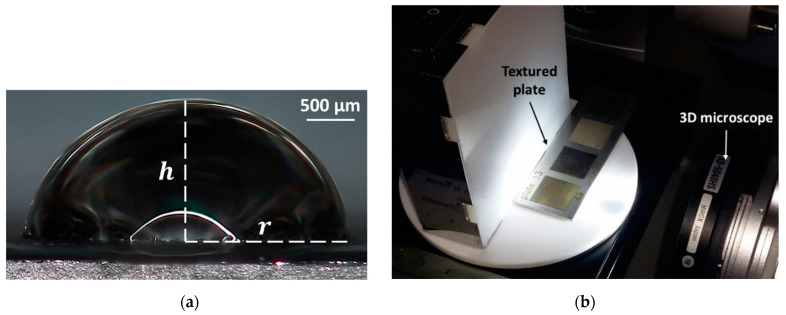
(**a**) Definition of the height (h) and radius (r) of the droplet for the evaluation of the contact angle through Equation (1); (**b**) Setup for the measurement of the contact angle by using a 3D microscope.

**Figure 3 materials-14-02379-f003:**
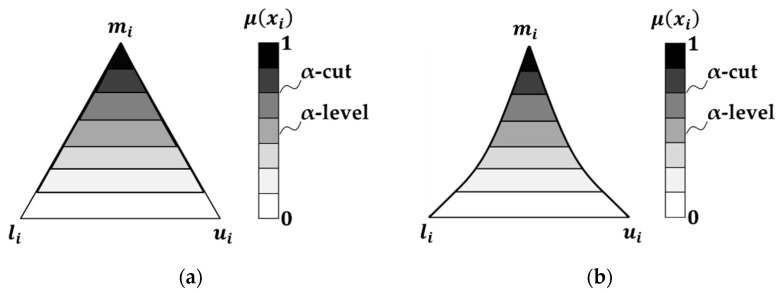
Representation of the fuzzy numbers: (**a**) triangular; (**b**) nonlinear. $\mu \left(x_{i}\right)$ is the membership function, represented as a gray shaded area, which defines the degree to which the parameter can take a certain value, i.e., as the membership function decreases, the level of uncertainty increases and the model accommodates a larger number of samples with lower membership level; $m_{i}$ is the modal value, i.e., the value of the coefficient calculated with the standard linear regression; $l_{i}$ and $u_{i}$ are the lower and the upper bounds of the fuzzy support.

**Figure 4 materials-14-02379-f004:**
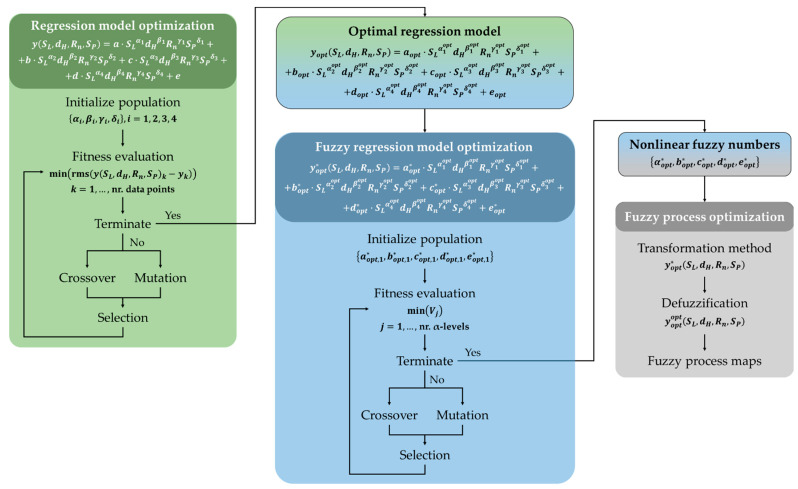
Flowchart of the algorithms adopted for the optimization of the regression models and the fuzzy numbers. The genetic algorithms are computed through four fundamental operations [53]: (i) initialization, with the definition of a set of powers for the regression model and of supports for the fuzzy numbers, and the fitness functions are applied to evaluate the fitness of this populations. If the stop condition is not met, they evolve into the next generations through (ii) crossover, which increases the variability of populations exchanging genes between two random chromosomes, (iii) mutation, which introduces random variation in the genome of some individuals, and (iv) selection, which allows transferring a defined number of chromosomes in the population to the next generation.

**Figure 5 materials-14-02379-f005:**
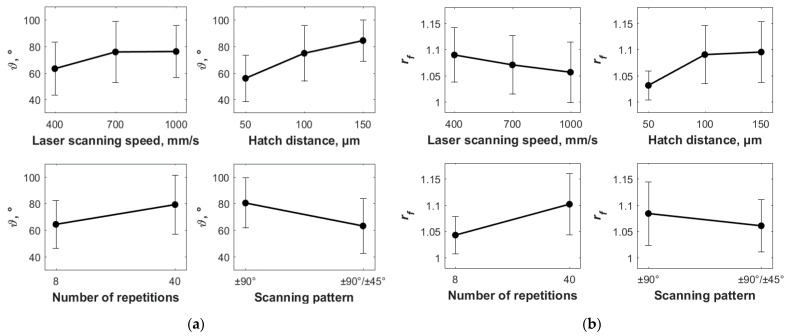
Experimental results in terms of (**a**) contact angle (ϑ) and (**b**) developed surface area ratio ($r_{f}$) about how they are affected by the operational process parameters varied in this investigation, i.e., laser scanning speed, hatch distance, number of repetitions, and scanning pattern (see Table 4 and Table 5). The black dots represent the mean values of the response variables, while the error bars represent the standard deviations.

**Figure 6 materials-14-02379-f006:**
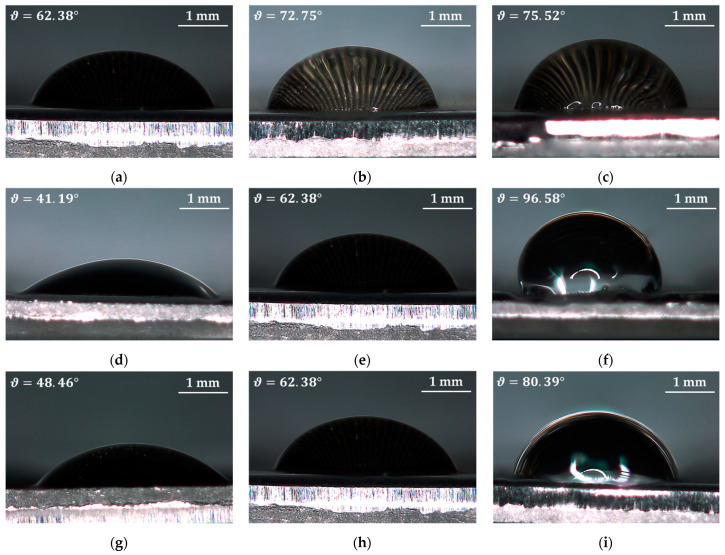
Drops images for the evaluation of the contact angles recorded with the setup shown in Figure 2 by adopting the 3D digital microscope: $d_{H}$ = 100 µm, $R_{n}$ = 8, $S_{P}$ = ±90°, and (**a**) $S_{L}$ = 400 mm/s, (**b**) $S_{L}$ = 700 mm/s, (**c**) $S_{L}$ = 1000 mm/s; $S_{L}$ = 400 mm/s, $R_{n}$ = 8, $S_{P}$ = ±90°, and (**d**) $d_{H}$ = 50 µm, (**e**) $d_{H}$ = 100 µm, (**f**) $d_{H}$ = 150 µm; $S_{L}$ = 400 mm/s, $d_{H}$ = 100 µm, $R_{n}$ = 8, and (**g**) $S_{P}$ = ±90°/±45°, (**h**) $S_{P}$ = ±90°; $S_{L}$ = 400 mm/s, $d_{H}$ = 100 µm, $S_{P}$ = ±90°, and (**h**) $R_{n}$ = 8, (**i**) $R_{n}$ = 40.

**Figure 7 materials-14-02379-f007:**
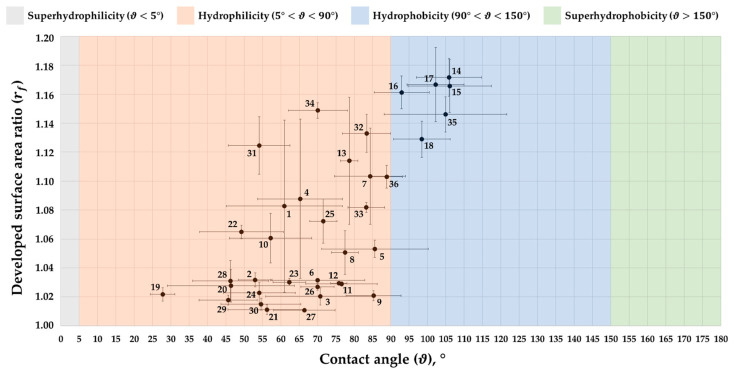
Wettability states observed during the experimental campaign. The numbers refer to the operational combinations of investigated parameters (see Table 4). The majority of the results are within the range of the hydrophilic state (5° < ϑ < 90°), while only six scenarios fall in the hydrophobic range (90° < ϑ < 150°), i.e., 14 to 18 and 35. For the details of the parameters’ values, please refer to Table 5.

**Figure 8 materials-14-02379-f008:**
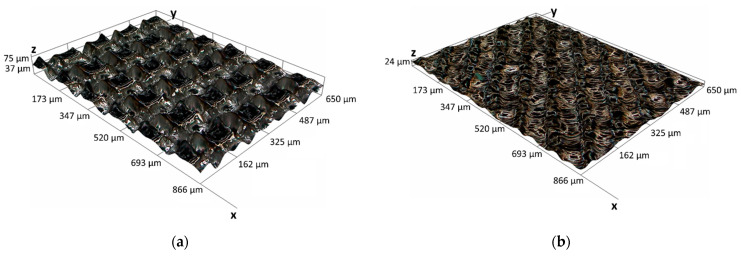
3D optical images of the surface topography for the experimental conditions (**a**) 18, i.e., $S_{L}$ = 1000 mm/s, $d_{H}$ = 150 µm,$\;R_{n}$ = 40, $S_{P}$ = ±90°, which allows the hydrophobic behaviour to take place; (**b**) 19, i.e., $S_{L}$ = 400 mm/s, $d_{H}$ = 50 µm,$\;R_{n}$ = 8, $S_{P}$ = ±90°/±45°, showing the surface topography that leads to the hydrophilic state.

**Figure 9 materials-14-02379-f009:**
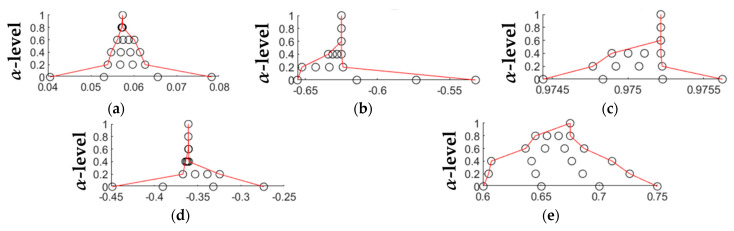
Nonlinear fuzzy numbers (**a**) $a_{opt}^{*}$, (**b**) $b_{opt}^{*}$, (**c**) $c_{opt}^{*}$, (**d**) $d_{opt}^{*}$, (**e**) $e_{opt}^{*}$, obtained starting from Equation (3) and optimized through the genetic algorithm presented in Section 2.2. The x-axes represent the support of the fuzzy numbers optimized at each α-level by taking into account a varying number of experimental data points according to the membership level, i.e., 100% at μ = 0, 80% at μ = 0.2, 60% at μ = 0.4; 40% at μ = 0.6, 20% at μ = 0.8, and 0% at μ = 1.

**Figure 10 materials-14-02379-f010:**
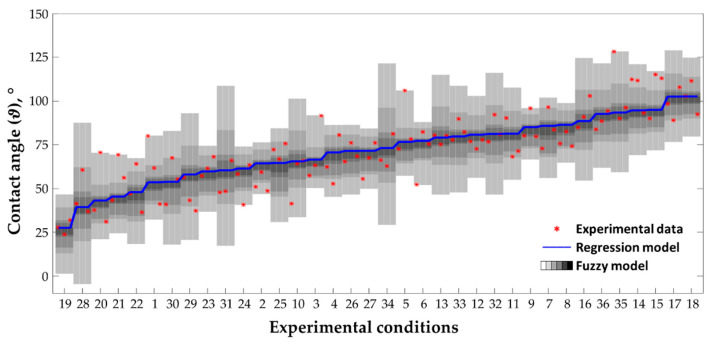
Fuzzy process map for the contact angle. The gray shaded area represents the membership function, which varies from white at μ = 0 while considering 100% of the experimental data points, to black at μ = 1 when the fuzzy number coincides with the empirical coefficients evaluated through standard linear regression. Each fuzzy band contains three data points (red asterisks) to which correspond a specific experimental condition, for a total of 36. The experimental tests are ordered for increasing values of the contact angle. For the experimental condition please refer to the parameters’ combinations reported in Table 5.

**Figure 11 materials-14-02379-f011:**
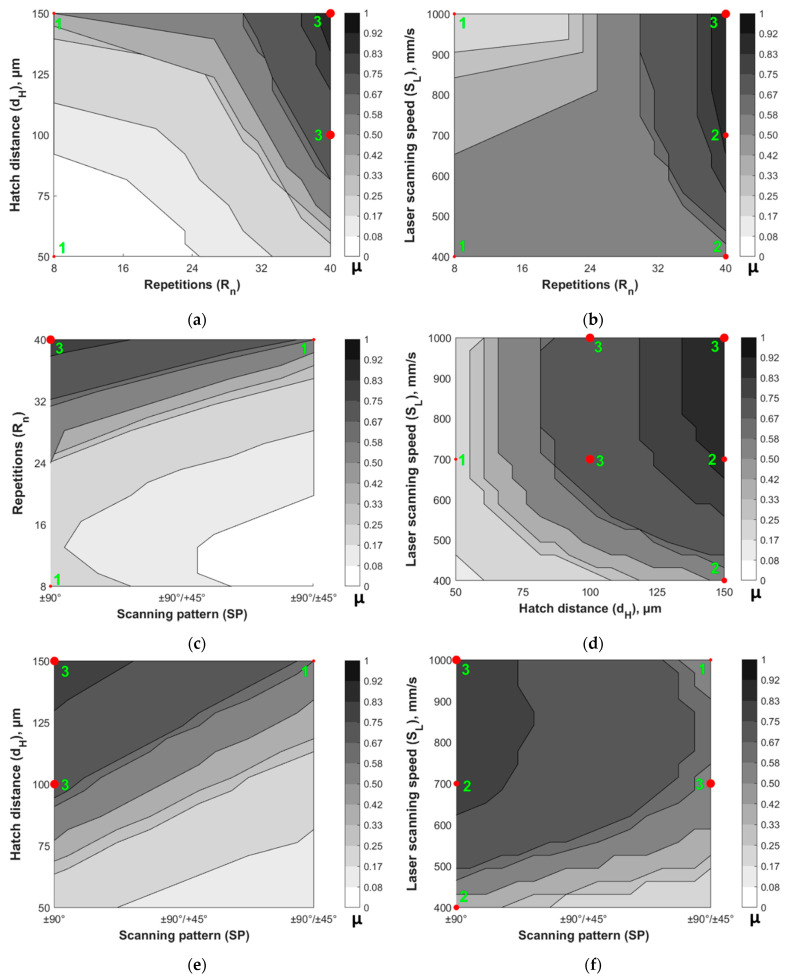
Fuzzy inverse maps for the obtainment of the hydrophobic state (90° < ϑ < 150°) by letting vary two parameters at a time while taking the others constant at their maximum levels: (**a**) varying $d_{H}$ and $R_{n}$, while $S_{L}$ = 1000 mm/s and SP = ±90°/±45°; (**b**) varying $S_{L}$ and $R_{n}$, while $d_{H}$ = 150 µm and SP = ±90°/±45°; (**c**) varying $R_{n}$ and SP, while $S_{L}$ = 1000 mm/s and $d_{H}$ = 150 µm; (**d**) varying $S_{L}$ and $d_{H}$, while $R_{n}$ = 40 and SP = ±90°/±45°; (**e**) varying $d_{H}$ and SP, while $S_{L}$ = 1000 mm/s and $R_{n}$ = 40; (**f**) varying $S_{L}$ and SP, while $d_{H}$ = 150 µm and $R_{n}$ = 40. The gray shaded area represents the membership function μ, ranging from 0 (white) to 1 (black) and describing the degree of belonging of a given sample to the model, i.e., from 100% at μ = 0, to 0% at μ = 1. The red dots and the green numbers represent the experimental occurrences.

**Table 1 materials-14-02379-t001:** Chemical composition of the starting material (AISI 304 stainless steel), as declared by the producer.

Element	Weight%	Element	Weight%
C	0.047	Mo	0.29
Cr	18.1	P	0.029
Ni	8.04	S	0.003
Mn	1.2	N	0.06
Si	0.48	-	-

**Table 2 materials-14-02379-t002:** Main mechanical properties of the starting material (AISI 304 stainless steel), as declared by the producer.

Properties	Values	Units
Elastic Modulus	193 ÷ 200	GPa
Ultimate Tensile Strength	505	MPa
Yield Tensile Strength	215	MPa
Hardness Rockwell B	70	-

**Table 3 materials-14-02379-t003:** Main characteristics of the Q-switched ns-pulsed Yb:YAG fibre laser system adopted to modify the surface topography of the AISI 304 samples.

Parameters	Values	Units
Wavelength	1.064	µm
Nominal average power	30	W
Maximum pulse energy	1	mJ
Maximum peak power	20	kW
Pulse frequency	30 ÷ 80	kHz
Pulse duration	50	ns
TEM	00	-
Focused spot diameter	~	µm
Working area	100×100	mm^2^

**Table 4 materials-14-02379-t004:** Experimental plan. The combination of three values for the laser scanning speed and the hatch distance, with two values for the number of repetitions and scanning pattern, results in 36 different experimental conditions. Each of them has been replicated three times for a total of 108 tests. The scanning pattern consisted in parallel lines along two (+90°, −90°) and four (+90°, −90°, +45°, −45°) directions. Therefore, the number of repetitions is an integer multiple of two.

Process Parameters	Values			Units
Pulse frequency	60			kHz
Laser scanning speed	400	700	1000	mm/s
Hatch distance	50	100	150	µm
Number of repetitions	8		40	-
Scanning pattern	±90		±90/±45	-

**Table 5 materials-14-02379-t005:** Operational parameters’ combinations investigated according to the experimental plan reported in Table 4: three values of laser scanning speed ($S_{L}$) and hatch distance ($d_{H}$), multiplied for two values of the number of repetitions ($R_{n}$) and scanning pattern (SP) results in 36 different experimental conditions.

Combinations	$S_{L}$, mm/s	$d_{H}$, µm	$R_{n}$	SP	Combinations	$S_{L}$, mm/s	$d_{H}$, µm	$R_{n}$	SP
1	400	50	8	±90°	19	400	50	8	±90°/±45°
2	700	50	8	±90°	20	700	50	8	±90°/±45°
3	1000	50	8	±90°	21	1000	50	8	±90°/±45°
4	400	100	8	±90°	22	400	100	8	±90°/±45°
5	700	100	8	±90°	23	700	100	8	±90°/±45°
6	1000	100	8	±90°	24	1000	100	8	±90°/±45°
7	400	150	8	±90°	25	400	150	8	±90°/±45°
8	700	150	8	±90°	26	700	150	8	±90°/±45°
9	1000	150	8	±90°	27	1000	150	8	±90°/±45°
10	400	50	40	±90°	28	400	50	40	±90°/±45°
11	700	50	40	±90°	29	700	50	40	±90°/±45°
12	1000	50	40	±90°	30	1000	50	40	±90°/±45°
13	400	100	40	±90°	31	400	100	40	±90°/±45°
14	700	100	40	±90°	32	700	100	40	±90°/±45°
15	1000	100	40	±90°	33	1000	100	40	±90°/±45°
16	400	150	40	±90°	34	400	150	40	±90°/±45°
17	700	150	40	±90°	35	700	150	40	±90°/±45°
18	1000	150	40	±90°	36	1000	150	40	±90°/±45°

**Table 6 materials-14-02379-t006:** ANOVA table for the contact angle (ϑ). The process parameters, i.e., laser scanning speed ($S_{L}$), hatch distance ($d_{H}$), number of repetitions ($R_{n}$), scanning pattern (SP), are considered statistically significant if *p*-value < 0.0.5, and F-value > 3.97 for DoF = 1, F-value > 3.12 for DoF = 2, or F-value > 2.50 for DoF = 4. DoF is the Degree of Freedom, Adj.SS is the adjusted sum of squares, Adj.MS is the adjusted mean sum of squares, Π is the contribution percentage (defined as the ratio between Adj.SS of the term and the total Adj.SS).

Source	DoF	Adj.SS	Adj.MS	F-Value	*p*-Value	Π (%)
$S_{L}$	2	3917.0	1958.48	14.05	0.000	7.90
$d_{H}$	2	15,094.8	7547.41	54.14	0.000	30.46
$R_{n}$	1	5987.8	5987.83	42.95	0.000	12.08
SP	1	8085.6	8085.56	58.00	0.000	16.32
$S_{L}\times d_{H}$	4	1353.6	338.39	2.43	0.056	2.73
$S_{L}\times R_{n}$	2	931.6	465.82	3.34	0.041	1.88
$S_{L}\times SP$	2	164.3	82.14	0.59	0.557	0.33
$d_{H}\times R_{n}$	2	937.5	468.75	3.36	0.040	1.89
$d_{H}\times SP$	2	464.4	232.18	1.67	0.196	0.94
$R_{n}\times SP$	1	16.0	16.01	0.11	0.736	0.03
*Error*	72	10,037.7	139.41	-	-	20.26
*Total*	107	49,552.4	-	-	-	-

**Table 7 materials-14-02379-t007:** ANOVA table for the developed surface area ratio ($r_{f}$). The process parameters, i.e., laser scanning speed ($S_{L}$), hatch distance ($d_{H}$), number of repetitions ($R_{n}$), scanning pattern (SP), are considered statistically significant if *p*-value < 0.0.5, and F-value > 3.97 for DoF = 1, F-value > 3.12 for DoF = 2, or F-value > 2.50 for DoF = 4. DoF is the Degree of Freedom, Adj.SS is the adjusted sum of squares, Adj.MS is the adjusted mean sum of squares, Π is the contribution percentage (defined as the ratio between Adj.SS of the term and the total Adj.SS).

Source	DoF	Adj.SS	Adj.MS	F-Value	*p*-Value	Π (%)
$S_{L}$	2	0.019595	0.009797	17.77	0.000	5.76
$d_{H}$	2	0.089902	0.044951	81.52	0.000	26.41
$R_{n}$	1	0.093332	0.093332	169.27	0.000	27.42
SP	1	0.014696	0.014696	26.65	0.000	4.32
$S_{L}\times d_{H}$	4	0.003771	0.000943	1.71	0.157	1.11
$S_{L}\times R_{n}$	2	0.008065	0.004032	7.31	0.001	2.37
$S_{L}\times SP$	2	0.001436	0.000718	1.30	0.278	0.42
$d_{H}\times R_{n}$	2	0.050656	0.025328	45.94	0.000	14.88
$d_{H}\times SP$	2	0.000273	0.000137	0.25	0.781	0.08
$R_{n}\times SP$	1	0.000088	0.000088	0.16	0.690	0.03
*Error*	72	0.039700	0.000551	-	-	11.66
*Total*	107	0.340417	-	-	-	-

**Table 8 materials-14-02379-t008:** Genetic algorithm-optimized powers ($\alpha_{i}^{opt}$, $\beta_{i}^{opt}$, $\gamma_{i}^{opt}$, $\delta_{i}^{opt}$, with i = 1, 2, 3, 4, for the first, second, third, and fourth term) and regression coefficients ($a_{opt}$, $b_{opt}$, $c_{opt}$, $d_{opt}$, $e_{opt}$, being the last the constant term) for the three output variables investigated, i.e., average surface roughness profile (Ra), contact angle (ϑ), and developed surface area ratio ($r_{f}$).

Terms	Values		
	Ra	ϑ	$r_{f}$
[$\alpha_{1}^{opt}$, $\beta_{1}^{opt}$, $\gamma_{1}^{opt}$, $\delta_{1}^{opt}$]	[0.5, −1, 0, 0]	[−0.5, 1, −1, 0]	[0, −1, −0.5, 1]
[$\alpha_{2}^{opt}$, $\beta_{2}^{opt}$, $\gamma_{2}^{opt}$, $\delta_{2}^{opt}$]	[−1, 1, 0.5, 0]	[0.5, 0, −1, 0.5]	[−0.5, 0.5, 0.5, 0]
[$\alpha_{3}^{opt}$, $\beta_{3}^{opt}$, $\gamma_{3}^{opt}$, $\delta_{3}^{opt}$]	[0.5, −0.5, 0, 0]	[1, 0, −0.5, 0.5]	[0.5, −1, 0.5, −0.5]
[$\alpha_{4}^{opt}$, $\beta_{4}^{opt}$, $\gamma_{4}^{opt}$, $\delta_{4}^{opt}$]	[−1, 0.5, −1, −0.5]	[0.5, −0.5, 0.5, 1]	[1, 0, 1, –1]
$a_{opt}$	−0.7933	0.6752	−0.0044
$b_{opt}$	−0.0825	0.0574	−0.0754
$c_{opt}$	1.3709	0.6250	−0.0892
$d_{opt}$	0.0420	0.9752	0.1616
$e_{opt}$	0.1513	−0.3607	1.0353

## Data Availability

Not applicable.

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
