# Peer review of "Soft Computing Techniques for Laser-Induced Surface Wettability Control"

_materials, 2021, doi:10.3390/ma14092379_

Round 1

Reviewer 1 Report

In their manuscript, Mr. Ponticelli et al. show their recent achievements on the fabrication of microtextured surfaces and the development of a regression model for describing the variation of the samples wettability. In particular, the model is based on fuzzy logic and is able to describe with high accuracy the correlation between process parameters and wetting response. The work shown in the manuscripit is valuable, presents a comprehensive theorethical part and reports the achievements in a correct and clear manner, also from the linguistic point of view. Neverthelss, some minor points can be further corrected and the overall form of the paper can be improved; therefore I suggest accepting the submission after minor revisions. In the following, some aspects that should be addressed in the revision phase.

- In the introduction, much is written about laser surface texturing, but very few information is giving on the computation side. For example, no information about the genetic algorythms (what they are and how they work) is given.

- In section 2, the pulse duration is not mentioned in the text. Please provide it.

- Is the maximum pulse energy really 1 J? (Table 3) 

- Reference missing in Line 259

- which role is playing the amount of experimental data gathered for "feeding" the algorythms? Is this sufficient, or lower than necessary? A clear comment on that would be beneficial in order to reproduce the reults in other labs or environments.

- In line 138 it mentioned that the frequency is an important parameter that can be changed, but according to the experiment setup it is always left at 60 kHz (line 144). Why is it mentioned at all?

- How are the patterned fields written? are they written sequentially or are they randomly sorted, as one would do for a DoE-approach?

- can the model be extended to other laser based techniques, as LIPSS or interference patterning? please comment on that in the conclusion part.

Author Response

The authors thank the Reviewer for his valuable comments and suggestions which are all accommodated in the revised manuscript improving the overall quality of the work.

Please see the attachment for the details.

Reviewer 2 Report

The authors conducted experimental and numerical modeling studies of laser-induced change of surface wettability on 304 stainless steel. The goal is to develop a regression model to map the input (laser processing parameters) with output (contact angle, surface morphology). The processing parameters are scanning speed, hatch distance, number of scans, and scan direction. This is a typical design-of-experiment study with novelty in the use of fuzzy logic. The experimental procedure is reasonable, and the promise of revealing both random and systemic errors using numerical modeling is appealing. The manuscript is clearly written. Figure 7 is particularly useful for other researchers. My comments are appended below.

  1. A major concern with this study is the universality of the modeling method. The genetic algorithm adopted here depends on initial guesses and there is no guarantee that everyone will reach the same global maximum (minimum), especially when there are differences in the laser processing conditions. The authors need to comment on this issue in the manuscript.

  1. The authors claim that the use of fuzzy logic helps with systematic errors, but it is not clear what these errors are. To convince the reader, the authors need to provide specific examples where there are hidden systematic errors which can indeed be revealed and compensated.

  1. It is hard for me to draw any sensible conclusion from Figure 5 because the amount of error is so large. Significant improvement is needed to reduce errors.

  1. Line 239: “… due to the simplification introduced by the model itself …” What is this simplification?

  1. Line 499: “Figure 11 suggests that the hydrophobic state can be promoted by adopting a high number of repetitions, greater than 24, while keeping the hatch distance over 75 μm, the laser scanning speed over 600 mm/s, and limiting the scanning pattern to ±90°.” The authors need to provide physical explanations as to why such parameters can result in hydrophobic surfaces.

  1. Typos: Line 48: “… can mitigate these limitations [20,21], but increasing the cost …”. Line 150: “The experimental tests were and performed …”. Line 259: Reference missing.

Author Response

(The authors gave the same response as above.)

Round 2

Reviewer 2 Report

The authors have addressed all my comments.